# Effects of Camu-Camu (*Myrciaria dubia*) Powder on the Physicochemical and Kinetic Parameters of Deteriorating Microorganisms and *Salmonella enterica* Subsp. *enterica* Serovar Typhimurium in Refrigerated Vacuum-Packed Ground Beef

Jorge Luiz da Silva [1,2,3], Vasco Cadavez [1], José M. Lorenzo [4], Eduardo Eustáquio de Souza Figueiredo [2,*] and Ursula Gonzales-Barron [1]

1   Centro de Investigação de Montanha (CIMO), Instituto Politécnico de Bragança (IPB),
    Campus of Santa Apolónia, 5300-253 Braganza, Portugal; jorge.silva@svc.ifmt.edu.br (J.L.d.S.);
    vcadavez@ipb.pt (V.C.); ubarron@ipb.pt (U.G.-B.)
2   Federal University of Mato Grosso (UFMT), Cuiabá, MT 78060-900, Brazil
3   Federal Institute of Education, Science and Technology of Mato Grosso (IFMT), Campo Verde,
    MT 78106-970, Brazil
4   Meat Technology Centre Foundation of Galicia, Technologic Park of the Galicia, San Cibrao das Viñas,
    32900 Ourense, Spain; jmlorenzo@ceteca.net
*   Correspondence: figueiredoeduardo@ufmt.br; Tel.: +55-65-3615-8589

**Abstract:** This study aims to evaluate the effects of camu-camu powder (CCP), Amazonian berry fruit with documented bioactive properties, physicochemical meat parameters, and the growth kinetics parameters of *S. enterica* ser. Typhimurium, psychrotrophic bacteria (PSY), and lactic acid bacteria (LAB) in vacuum-packed ground beef. Batches of ground beef were mixed with 0.0%, 2.0%, 3.5%, and 5.0% CCP (*w*/*w*), vacuum-packed as 10 g portions, and stored at 5 °C for 16 days. Centesimal composition analyses (only on the initial day), pH, TBARS, and color were quantified on storage days 1, 7, and 15, while PSY and LAB were counted on days 0, 3, 6, 9, 13, and 16. Another experiment was conducted with the same camu-camu doses by inoculating *S. enterica* ser. Typhimurium microbial kinetic curves were modeled by the Huang growth and Weibull decay models. CCP decreased TBARS in beef from 0.477 to 0.189 mg MDA·kg$^{-1}$. No significant differences in meat pH between treated and control samples were observed on day 15. CCP addition caused color changes, with color a* value decreases (from 14.45 to 13.44) and color b* value increases (from 17.41 to 21.25), while color L* was not affected. Higher CCP doses caused progressive LAB growth inhibition from 0.596 to 0.349 log CFU·day$^{-1}$ at 2.0% and 5.0% CCP, respectively. Similarly, PSY growth rates in the treated group were lower (0.79–0.91 log CFU·day$^{-1}$) compared to the control (1.21 log CFU·day$^{-1}$). CCP addition at any of the investigated doses produced a steeper *S. enterica* ser. Typhimurium inactivation during the first cold storage day, represented by Weibull's concavity $\alpha$ shape parameter, ranged from 0.37 to 0.51, in contrast to 1.24 for the control. At the end of the experiment, however, *S. enterica* ser. Typhimurium counts in beef containing CCP were not significantly different ($p < 0.05$) from the control. Although CCP affects bacterial kinetics, it does not protect ground beef against spoilage bacteria and *Salmonella* to the same degree it does against lipid peroxidation.

**Keywords:** camu-camu powder; meat; pH; *S. enterica* ser. Typhimurium; spoilage bacteria; TBARS

## 1. Introduction

The camu-camu (*Myrciaria dubia*) is a reddish fruit color found in trees of typical Amazon forest wild species [1–4]. This fruit exhibits a high phytochemical profile, conferring nutritional and functional values to this fruit and economic importance. Camu-camu is essential for the food, pharmaceutical, and cosmetics industries due to its bioactive,

antioxidant, anti-inflammatory, and antimicrobial properties while also containing high vitamin C, carotenoid, and phenolic compound contents [3–10].

The beef industry has attempted to use natural compounds displaying functional and antimicrobial properties to formulate products to promote health and food safety [11]. The presence of *Salmonella* in beef is a relevant concern among biological risks, as this pathogen is involved with the most food-borne disease outbreaks [12]. Thus, alternatives such as natural compounds to control *Salmonella* in foodstuffs are increasingly required.

Previous assessments concerning camu-camu powder (CCP) have indicated phenolic compounds such as flavonoids, anthocyanins, ellagic acid derivatives, ellagitannins, gallic acid derivatives, and proanthocyanidins in the pulp, seed, and fruit peel [4,13]. Among phenolic compounds, CCP contains flavones, quercetin, and naringenin, which play a role in microorganism inhibition [14]. Both fresh and dry camu-camu exhibit the highest antioxidant capacity and polyphenol contents among 18 native non-traditional tropical Brazilian fruits [15].

Alcoholic camu-camu seed and pulp extracts exhibit high anti-bactericidal activity against *Streptococcus mutans* and *Streptococcus sanguinis* [16], while aqueous camu-camu extract fractions, rich in phenolic compounds, display inhibitory action against *Staphylococcus aureus* [3]. In another study, CCP methanol/water extracts (70:30 *v/v*) showed inhibitory effects against *S. aureus.* However, no effects were observed against other microorganisms, such as *Escherichia coli, Enterobacter aerogenes, Listeria monocytogenes, S. enterica* ser. Typhimurium or *Salmonella enterica ser.* Enteritidis [6]. Some studies have evaluated the antimicrobial properties of camu-camu extracts, although CCP effects in meat have not yet been evaluated.

In this context, the present study aimed to evaluate CCP effects on physicochemical meat properties and the behavior of psychrotrophic bacteria, lactic acid bacteria, and *S. enterica* ser. Typhimurium in vacuum-packed ground beef during cold storage. So, these results can be used in meat industries to use one natural compound in the processes.

## 2. Material and Methods

This study comprised two experiments. In the first experiment, physicochemical characteristics and spoilage microorganisms were quantified along time in non-inoculated ground beef samples to evaluate if CCP impacts the shelf-life and quality attributes of ground meat. The second experiment consisted of a challenge study where *S. enterica* ser. Typhimurium was inoculated in ground beef containing CCP to characterize its kinetic parameters of behavior. Beef samples were obtained from a slaughterhouse located in Braganza, Portugal, while raw whole camu-camu powder (CCP) was purchased from Bio-Aurora Industry (Lima, Peru). The proximate CCP composition, informed by the industry, was as follows: moisture 6.08%, fat 0.06%, protein 1.64%, ashes 1.52%, carbohydrates 90.73%, fiber 0.81% with 15.02% titratable acidity (citric acid), and 7397 mg/100 g vitamin C.

### 2.1. Sample Preparation

A total of four batches (420 g each) of ground beef were mixed with CCP at 0.0%, 2.0%, 3.5%, and 5.0% (*w/w*), respectively, using a semi-professional mixer (model Artisan 5KSM125, KitchenAid, Benton Harbor, MI, USA) for 5 min at speed 4 (~100 rpm). Ten-gram portions were then vacuum packed in transparent gas-tight polyamide and polyethylene bags (grades vacuum bags, Orved®, Musile di Piave, Venice, Italy, with the permeability of $84 \pm 4.20$ cc/m$^2$/24 h/atm for O$_2$, $361 \pm 18.05$ cc/m$^2$/24 h/atm for CO$_2$, $22 \pm 1.10$ cc/ m$^2$/24 h/atm for N$_2$ and $9.0 \pm 0.45$ cc/m$^2$/24 h/atm for H$_2$O and a density of $\pm 100$ μm) using a vacuum sealer (Silvercrest SFS 110B2, Bochum, Germany), and stored at 5 °C in a cooling chamber (Froztec, Miramar, FL, USA).

#### 2.1.1. Physicochemical Analyses

Three samples (10 g portions) per treatment were analyzed to determine the physicochemical parameters (pH, color, and lipid oxidation) on days 1, 7, and 15. The proximate

composition (moisture, lipids, protein, and ash) was determined on day 0, per treatment, according to AOAC [17] and ISO [18] in duplicate subsamples, while physicochemical parameters were determined in five subsamples. The pH values were determined using a potentiometer (HI 9913, Hanna Instruments, Eibar, Spain) and color (L*, a*, b*) by a Minolta colorimeter (Konica Minolta CM-600d, Osaka, Japan). Lipid oxidation was determined through the thiobarbituric acid reaction (TBARS) according to Vyncke [19]. Briefly, a 10 g portion was added to 97.5 mL of distilled water containing 2.5 mL of 4 N HCl, homogenized and distilled. Then, 5 mL of the distillate was added to 5 mL of a 0.02 M TBA solution and heated in a water bath at 100 °C for 35 min. Measurements were carried out using a Specord 200 spectrophotometer (Analytik Jena AG, Jena, Germany) at 528 nm, and the results were expressed as mg of malondialdehyde (MDA)/kg.

### 2.1.2. Microbiological Analyses (PSY and LAB)

Total psychrotrophic and lactic acid bacteria counts were performed on storage days 0, 3, 6, 9, 13, and 16. Sample (10 g) was added to 90 mL of buffered peptone water 2% (Oxoid, Cheshire, UK) and homogenized for 60 s (Stomacher 400, Seward, West Sussex, UK), followed by serial dilutions preparations.

According to APHA [20], psychrotrophic plate counts were performed where 1 mL of appropriate dilutions, in duplicate, were pour-plated into PCA agar (Liofilchem® s.r.l., Roseto Degli Abruzzi, Italy) and incubated at 7 °C for 10 days. For lactic acid bacteria, 1 mL of appropriate dilutions were pour-plated in duplicate in MRS agar (Liofilchem® s.r.l., Roseto Degli Abruzzi, Italy) supplemented with Tween 80 (Liofilchem® s.r.l., Roseto Degli Abruzzi, Italy), and overlaid with MRS agar. The plates were then incubated in anaerobic conditions at 30 °C for 48 h. Typical colonies were counted with a colony counter (Digital S®, J.P. Selecta S.A., Barcelona, Spain), and the data were converted to $\log_{10}$ CFU·$g^{-1}$.

### 2.2. S. enterica ser. Typhimurium Behaviour in Vacuum-Packed Ground Beef Containing CCP

One loop of *S. enterica* ser. Typhimurium ATCC 14028 strain stored in a cryo-vial at −80 °C was transferred to TSB broth (Liofilchem® s.r.l., Roseto Degli Abruzzi, Italy) and incubated 35 °C for 24 h. Subsequently, a loopful was streaked on XLD agar (Liofilchem® s.r.l., Roseto Degli Abruzzi, Italy) and incubated at 35 °C for 24 h to obtain isolated colonies. One isolated colony was then transferred to a new TSB broth tube and further incubated at 35 °C for 24 h. A loopful was finally transferred to TSB broth (Liofilchem® s.r.l., Roseto Degli Abruzzi, Italy) and incubated at 8 °C (slow growth condition) until reaching the early stationary phase, determined with the aid of a previously constructed calibration curve determined using a Specord 200 spectrophotometer (Analytik Jena AG, Jena, Germany) at 600 nm.

Batches (200 g) of ground beef were mixed with CCP at 0.0%, 2.0%, 3.5%, or 5.0% (*w/w*) using a semi-professional mixer (model Artisan 5KSM125, KitchenAid, Benton Harbor, MI, USA) for 5 min at speed 4 (~100 rpm). An appropriate amount of the refrigerated inoculum was diluted in physiological water (5 mL) added to the bulk ground beef to target an *S. enterica* ser. Typhimurium concentration of 5 log CFU/$g^{-1}$. The ground beef was further mixed for 7 min. Three portions of ten-gram each were then vacuum packed in transparent gas-tight polyamide and polyethylene bags (grades vacuum bags, Orved®, Spain, with the permeability of $84 \pm 4.20$ cc/$m^2$/24 h/atm for $O_2$, $361 \pm 18.05$ cc/$m^2$/24 h/atm for $CO_2$, $22 \pm 1.10$ cc/$m^2$/24 h/atm for $N_2$ and $9.0 \pm 0.45$ cc/$m^2$/24 h/atm for $H_2O$ and density of $\pm 100\mu$m) using a vacuum sealer (Silvercrest SFS 110B2, Bochum, Germany), and stored at 5 °C.

*S. enterica* ser. Typhimurium was enumerated on storage days 0, 3, 6, 9, 12, and 15 in three samples analyzed per treatment and time point. Ten-gram samples were then mixed with 90 mL of buffered peptone water 2% (Oxoid, Cheshire, UK) and homogenized for 60 s (Stomacher 400, Seward, West Sussex, UK). Serial dilutions were performed, and 0.1 mL of the appropriate dilutions were inoculated, in duplicate, onto plates containing Hektoen Enteric Agar (Liofilchem® s.r.l., Roseto Degli Abruzzi, Italy) and incubated at 35 °C for

24 h. Typical colonies were counted with a colony counter (Digital S®, J.P. Selecta S.A., Barcelona, Spain), and the data were converted to $\log_{10}$ CFU·g$^{-1}$.

### 2.3. Statistical Analyses

The physicochemical data, namely pH, L*, a*, b*, TBARS, and proximate analysis, underwent an analysis of variance and Dunnett's comparison of means tests to determine potential differences CCP levels at a significance probability level of 0.05. All statistical analyses were carried out using the R software [21].

Each of the experimental psychrotrophic bacteria and LAB growth curves was modeled by adjusting the integrated form of the Huang primary model (Equation (1)) proposed for a constant environmental condition [22], as follows:

$$Y(t) = Y_0 + Y_{max} - ln\{\exp(Y_0) + (\exp(Y_{max}) - \exp(Y_0)) \times \exp(-\mu_{max}B(t))\}$$
$$B(t) = t + \frac{1}{\alpha}ln\frac{1+\exp(-\alpha(t-\lambda))}{1+\exp(\alpha\lambda)} \tag{1}$$

where: $Y_0$, $Y_{max}$, and $Y$ represent the natural logarithms of microbial concentrations at an initial time point ($t = 0$), maximum population and actual time $t$, respectively; $\mu_{max}$ accounts for a maximum specific growth rate (day$^{-1}$); $\lambda$ is the delay interval (or lag time) of a curve depicting microbial behavior through time (day); $\alpha$ is the coefficient that accounts for the lag phase shift (set to 4, as recommended by Huang [23]); and t is the time interval. Since LAB curves did not exhibit any lag phase, $\lambda$ was set to zero in Equation (1) when adjusting the Huang model.

*S. enterica* ser. Typhimurium decay behavior was modeled by a three-parameter modified Weibull equation (Equation (2)), defined as

$$Y(t) = Y_0 - \left(\frac{t}{\chi}\right)^{\beta} \tag{2}$$

where the scale and shape parameters of the underlying Weibull distribution are $\chi$ and $\beta$, respectively. In shape parameter $\beta > 1$, convex curves are obtained, and for $\beta < 1$, concave curves are represented. Although the Weibull model is basically empirical, van Boekel [24] suggested that $\beta < 1$ presumes that the surviving microorganisms at any point in the inactivation curve display the capacity to adapt to the applied stress, whereas $\beta > 1$ indicates that the remaining cells become increasingly susceptible to heat. The parameter $\chi$ is called scale parameter (a characteristic time). $Y$ is defined as above, and the parameter $Y_0$ represents the natural logarithm of the initial microbial concentration, and t represents the time.

## 3. Results

### 3.1. Physicochemical Characterization of Vacuum-Packed and Refrigerated Beef Containing Camu-Camu Powder

The centesimal composition of the ground beef samples presented mean values of 68.13%, 3.79%, 23.71%, and 1.31% for moisture, lipids, proteins, and ashes, respectively. Table 1 presents the proximate composition of the meat samples subjected to CCP treatments.

**Table 1.** Proximate composition (% wb) of ground beef control and treatments with the addition of camu-camu powder as determined on the initial day of the experiment.

| CCP (%) | n | Moisture (%) | Lipids (%) | Protein (%) | Ashes (%) |
|---------|---|--------------|------------|-------------|-----------|
| 0.0% | 5 | 71.2 | 3.4 | 24.1 | 1.3 |
| 2.0% | 5 | 68.7 | 4.3 | 23.7 | 1.3 |
| 3.5% | 5 | 67.5 | 4.3 | 23.7 | 1.3 |
| 5.0% | 5 | 66.8 | 3.3 | 23.6 | 1.4 |

CCP = camu-camu powder; n = number of samples analyzed.

Concerning physicochemical parameters, significant effects were observed as a function of CCP addition, storage time (days), and the interactions of these factors. pH values varied ($p < 0.05$) as a function of CCP addition and time. However, higher CCP concentrations resulted in increases in pH values of 5.4 (0.0%) to 5.6 (5.0%). Considering storage time, pH values decreased from 5.9 to 5.3 between days 1 and 15, respectively. Regarding color changes, significant effects on L*, a*, b* were observed with CCP addition, while significant effects during the storage time were observed only for a* and b* (Table 2).

**Table 2.** Physicochemical parameters determined in vacuum-packed ground beef containing different levels of camu-camu powder and stored at 5 °C.

| Main Effects | Parameters | | | | |
|---|---|---|---|---|---|
| | pH | L* | a* | b* | TBARs [mg MDA/kg] |
| CCP (%) | *** | *** | *** | *** | *** |
| 0.0 | 5.476 [a] ± 0.051 | 49.11 [c] ± 0.544 | 14.45 [b] ± 0.401 | 17.41 [a] ± 0.354 | 0.573 [c] ± 0.065 |
| 2.0 | 5.576 [b] ± 0.069 | 44.34 [b] ± 0.260 | 14.44 [b] ± 0.306 | 20.08 [b] ± 0.180 | 0.464 [bc] ± 0.069 |
| 3.5 | 5.640 [c] ± 0.085 | 43.03 [ab] ± 0.333 | 13.96 [ab] ± 0.229 | 21.23 [c] ± 0.189 | 0.289 [ab] ± 0.048 |
| 5.0 | 5.662 [c] ± 0.069 | 41.92 [a] ± 0.222 | 13.44 [a] ± 0.212 | 21.25 [c] ± 0.215 | 0.207 [a] ± 0.048 |
| Time (days) | *** | NS | *** | *** | *** |
| 1 | 5.933 [c] ± 0.029 | 44.24 [a] ± 0.645 | 14.28 [b] ± 0.281 | 19.8 [a] ± 0.541 | 0.066 [a] ± 0.017 |
| 7 | 5.488 [b] ± 0.026 | 44.82 [a] ± 0.831 | 14.48 [b] ± 0.222 | 20.6 [b] ± 0.296 | 0.403 [b] ± 0.044 |
| 15 | 5.344 [a] ± 0.010 | 44.75 [a] ± 0.560 | 13.46 [a] ± 0.251 | 19.6 [a] ± 0.334 | 0.501 [c] ± 0.050 |
| CCP × Time | *** | NS | *** | ** | ** |

[a,b,c] Means ± standard error followed by the same letter in the columns do not differ according to Dunnett's test at 5% significant: NS = non-significant. *** and ** = 0.001% and 0.01% of significance, respectively, by ANOVA test.

Camu-camu powder addition promoted decreases ($p < 0.05$) in meat lipid oxidation with values from 0.573 mg malondialdehyde (MA)/Kg (control group) to 0.207 mg malondialdehyde (MA)/kg (5.0% CCP) (Table 2). Interactions between CCP addition and storage time were also observed for TBARS values ($p < 0.05$) (Table 2). Higher decreases in lipid oxidation were noted for the 5.0% CCP concentration during nine storage days.

Considering the interaction between CCP concentration and time, pH decreases ($p < 0.05$) were observed from the first day until the end of the experiment for all treatments, while the CCP addition caused lipid oxidation declines, resulting in the lowest TBARS concentrations in samples containing 5.0% CCP. A progressive lipid oxidation inhibition as a function of increasing CCP concentrations was also observed (Figure 1).

Concerning color characteristics, interactions between CCP addition and time resulted in meat color changes, with decreases in L* and a*, and increases in b* values (Table 2). The control treatment presented higher L* values (49.11) compared to the CCP samples (44.34–41.92) ($p < 0.05$). Decreased ($p < 0.05$) a* values from 14.45 (control) to 13.44 (5.0% CCP) were observed, causing a red color reduction in the meat samples. Concerning b* (yellow hue), CCP addition caused increases from 17.4 (control) to 20.08 up to 21.25 in the CCP-containing samples (Figure 2).

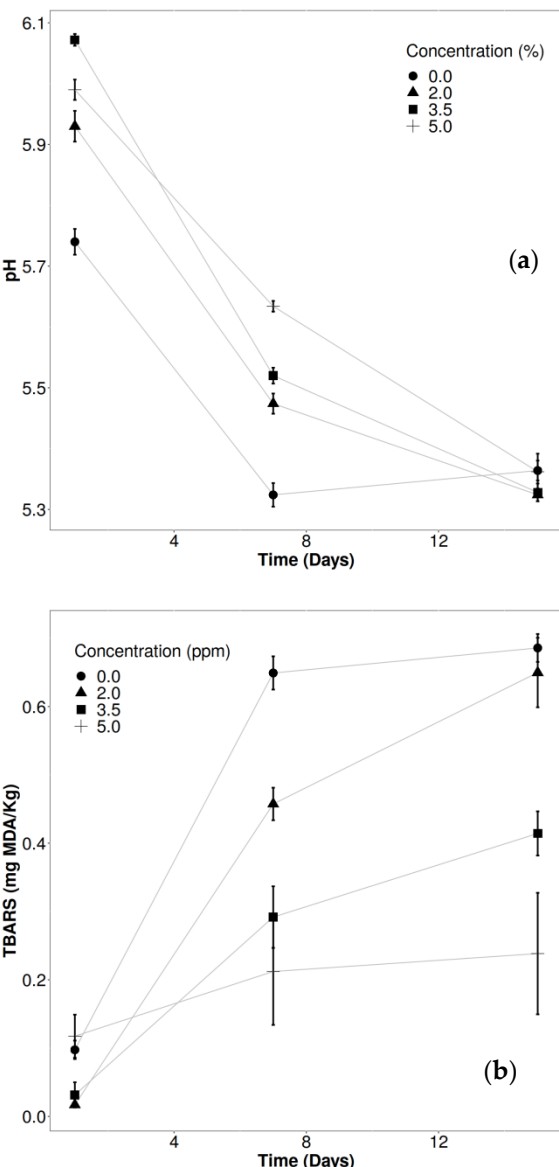

**Figure 1.** pH (**a**) and TBARS (**b**) of vacuum-packed ground beef containing different camu-camu powder levels stored at 5 °C.

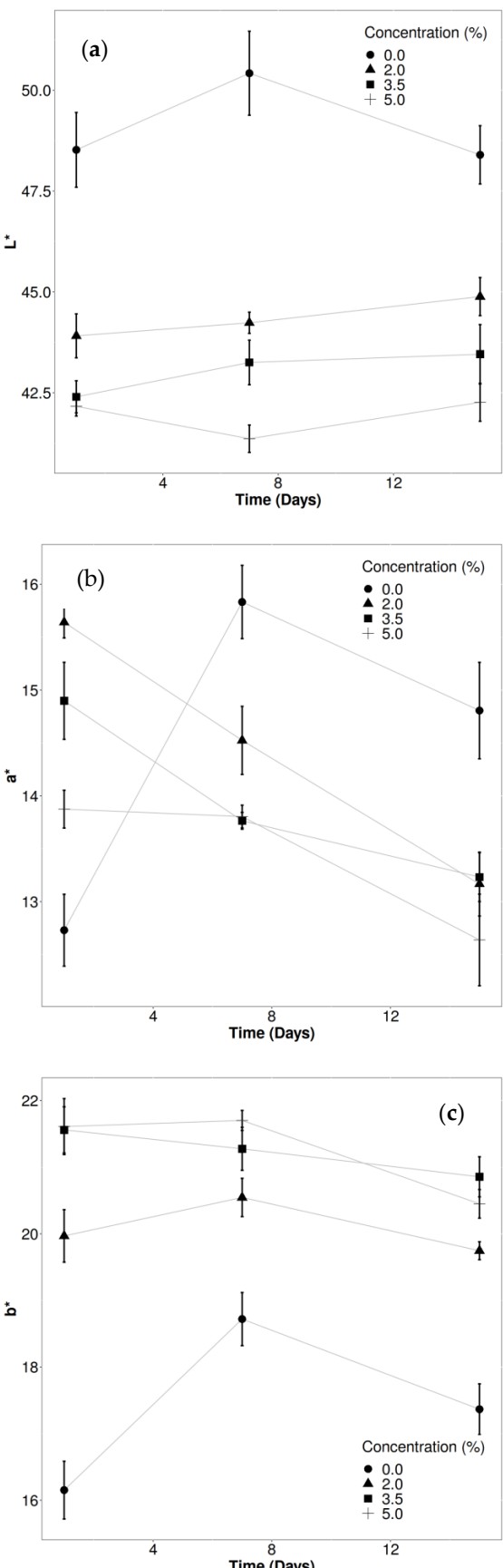

**Figure 2.** Colors, L* (**a**), a* (**b**), b* (**c**) of vacuum-packed ground beef containing different camu-camu powder levels stored at 5 °C.

### 3.2. Behavior of Deteriorating Microorganisms and S. enterica ser. Typhimurium in Vacuum-Packed Ground Beef

Kinetic parameter effects on both spoilage bacteria and *Salmonella* were caused by CCP addition. Although CCP additions decreased LAB kinetic parameters, such as $\mu_{max}$, microbial growth did not show changes, with decreased initial concentrations and increased final concentrations (Table 3).

**Table 3.** Kinetic parameters of the Huang growth model describing lactic acid bacteria concentrations in vacuum-packed ground beef containing different camu-camu powder levels stored at 5 °C.

| Model Parameters [1] | Camu-Camu Powder Concentration | | | |
|---|---|---|---|---|
| | 0.0% | 2.0% | 3.5% | 5.0% |
| $Y_0$ | 3.696 ± 0.227 *[2] | 3.655 ± 0.277 * | 3.584 ± 0.106 * | 3.851 ± 0.375 * |
| $Y_{max}$ | 6.941 ± 0.175 * | 7.019 ± 0.186 * | 6.985 ± 0.091 * | 7.601 ± 0.532 * |
| $\mu_{max}$ | 0.490 ± 0.079 * | 0.596 ± 0.112 * | 0.451 ± 0.033 * | 0.349 ± 0.092 * |

[1] $Y_0$: initial counts (log CFU/g); $Y_{max}$: final counts (log CFU/g); $\mu_{max}$: maximum growth rate (day$^{-1}$) (these parameters were expressed as means and standard error). [2] Asterisks represent the significance of the estimated parameter at $p < 0.05$.

CCP addition caused a psychrotrophic bacteria lag phase increase from 7.955 to 8.156 days in the controls and 5.0% in the treated groups, respectively (Table 4). However, no significant initial and final concentration effects were observed.

**Table 4.** Kinetic parameters of the Huang growth model describing psychrotrophic bacteria concentrations in vacuum-packed ground beef containing different camu-camu powder levels stored at 5 °C.

| Model Parameters [1] | Camu-Camu Powder Concentration | | | |
|---|---|---|---|---|
| | 0.0% | 2.0% | 3.5% | 5.0% |
| $Y_0$ | 5.092 ± 0.016 *[2] | 5.060 ± 0.022 * | 4.910 ± 0.080 * | 4.845 ± 0.158 * |
| $\lambda$ | 7.955 ± 0.308 * | 7.438 ± 0.270 * | 7.599 ± 0.917 * | 8.156 ± 0.804 * |
| $Y_{max}$ | 7.871 ± 0.030 * | 7.470 ± 0.046 * | 7.309 ± 0.166 * | 7.579 ± 0.315 * |
| $\mu_{max}$ | 1.210 ± 0.350 * | 0.788 ± 0.133 * | 0.828 ± 0.527 * | 0.912 ± 0.694 * |

[1] $Y_0$: initial counts (log CFU/g); $\lambda$: lag phase duration (day); $Y_{max}$: final counts (log CFU/g); $\mu_{max}$: maximum growth rate (day$^{-1}$) (these parameters were expressed as means and standard error). [2] Asterisks represent the significance of the estimated parameter at $p < 0.05$.

The results for other kinetic parameters did not present a statistical difference with initial concentration, and the maximum concentration at final storage time was between 7.30 to 7.87 log CFU/g, indicating the start of deterioration. Although no statistical effect, it should be highlighted the $\mu_{max}$ difference between the control group and treated groups showing kinetic action.

Significant effects ($p < 0.05$) were observed concerning *Salmonella* concentrations and kinetics in vacuum-packed ground beef stored at 5 °C (Table 5). Increasing in $\chi$ parameters, and the decrease in $\beta$ parameters indicate higher declines in pathogen concentrations on the first days of cold storage.

**Table 5.** Kinetic parameters of the Weibull decay model describing *S. enterica* ser. Typhimurium behavior in vacuum-packed ground beef containing different camu-camu powder levels stored at 5 °C.

| Model Parameters [1] | Camu-Camu Powder Concentration | | | |
|---|---|---|---|---|
| | **0.0%** | **2.0%** | **3.5%** | **5.0%** |
| $Y_0$ | 5.121 ± 0.048 *[2] | 5.046 ± 0.072 * | 5.091 ± 0.028 * | 5.089 ± 0.040 * |
| $\chi$ | 13.65 ± 1.710 * | 13.61 ± 5.192 * | 26.90 ± 7.319 * | 40.32 ± 20.34 [ns] |
| $\beta$ | 1.247 ± 0.442 * | 0.519 ± 0.240 * | 0.433 ± 0.110 * | 0.359 ± 0.103 * |

[1] $Y_0$: initial counts (log CFU/g); $\chi$: scale parameter (day$^{-1}$); $\beta$: shape parameter (dimensionless) (these parameters were expressed as means and standard error). [2] Asterisks represents the significance of the estimated parameter at $p < 0.05$; ns: non-significant.

No variations were observed in final pathogen concentrations, with values ranging from 4.61 to 4.79 log CFU/g. Therefore, the decreased bacterial concentrations observed in the present study were due to the low storage temperature (Figure 3).

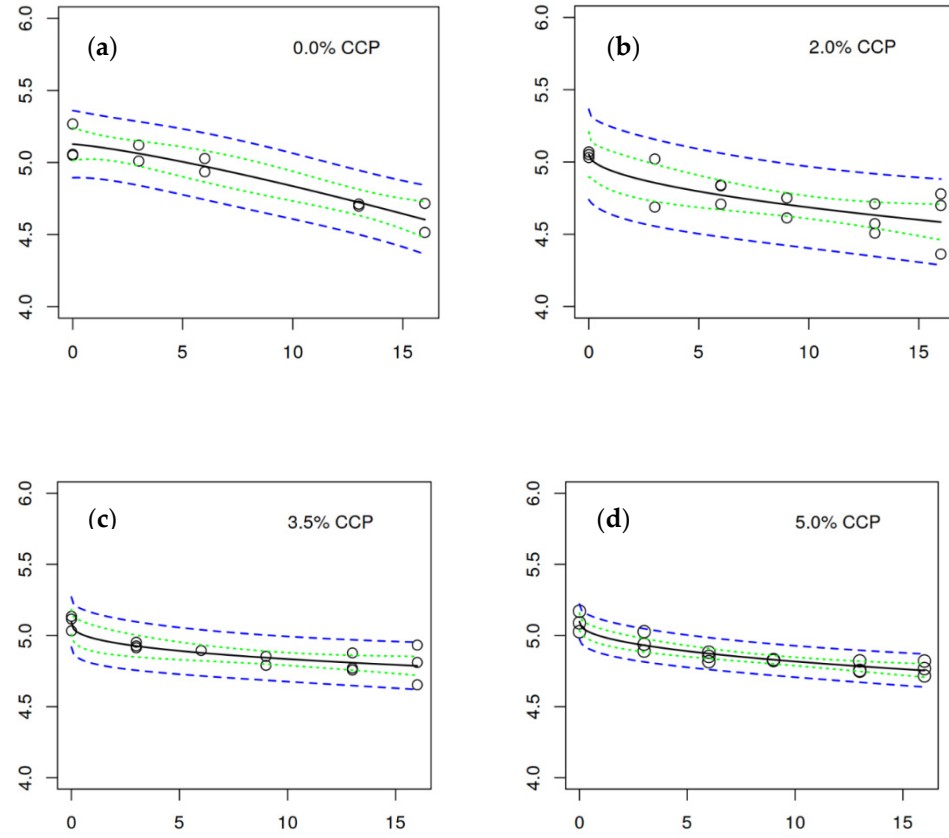

**Figure 3.** Weibull model fitting to experimental survival curves of *S. enterica* ser. Typhimurium in vacuum-packed ground beef containing different camu-camu powder (CCP) levels stored at 5 °C: 0.0% (**a**), 2.0% (**b**), 3.5% (**c**) and 5.0% (**d**). The 95% confidence intervals are displayed as green lines and the prediction intervals as blue lines. *X*-axis: Time (day); *y*-axis: Counts (log CFU/g).

## 4. Discussion

The addition of CCP caused moisture variations between 71.2% (control) and 66.78% (5.0%), while for lipids these values were between 3.34% (5.0% CCP) and 4.28% (2.0% CCP). Protein and ashes presented values between 23.70% and 24.13% and 1.27% and 1.37% respectively. Proximate composition alterations are not unexpected since camu-camu berries, apart from water, are mainly composed of carbohydrates (4.84% wb), crude fiber (0.56% wb), and several minerals such as potassium (87.0 mg/100 g, phosphorous (18.2 mg/100 g),

sulphate (14.7 mg/100 g), calcium (9.1 mg/100 g), magnesium (7.4 mg/100 g), cobalt (1.2 mg/100 g), and iron (0.42 mg/100 g), among others [9].

In one experiment, the addition of 100 ppm of camu-camu peel and seed extracts led to increases in pH values from 6.01 to 6.16 in ground-lamb meat stored at 4 °C for nine days under normal atmosphere [25], which was also observed herein (Table 2). Another study reported that pH values increased in lamb meat after 10 days of refrigeration due to medium alkalinization by bacteria and endogenous enzymatic protein degradation [26]. In the present study, bacterial spoilage growth (PSY and LAB) was observed during refrigerated storage of the CCP containing- vacuum-packed beef.

Lipid oxidation decreases were observed herein with CCP addition. In another experiment, camu-camu extracts affected ($p < 0.05$) lipid oxidation values in chilled ground lamb meat treated with camu-camu peel and seed extracts and stored at 4 °C for 9 days under a normal atmosphere, where the control group presented 0.108 mg MDA/kg while treat groups presented 0.079 and 0.068 mg MDA/kg for peel and seed extracts respectively [25]. The camu-camu extracts promoted food stability, which may be associated with phenolic compounds present in camu-camu fruits. In the present survey, TBARS decreases were also observed in the meat containing CCP, indicating that phenolic compounds are available in both the extract and powdered forms of the fruit.

Lipid peroxidation decreases between 24% and 86% ($p \leq 0.05$) have been reported in egg homogenates due to the use of aqueous and ethyl alcohol camu-camu seed extracts in studies using DPPH (2,2-diphenyl-1-picrylhydrazyl) radical activity and FCRC (Follin-Ciocalteu reducing capacity) tests [27]. Lipid oxidation inhibition was correlated with total phenolic contents (r = 0.78), total flavonoid contents (r = 0.67), and condensed tannin contents (r = 0.68), and may be an effect of reaction mechanisms, resulting in greater antioxidant activity [27]. Another study reported camu-camu flour (CCF) and camu-camu pulp powder (CCPP) antioxidant capacities, as both the products contain both vitamin C and phenolic compounds [13]. In a DPPH assay, CCF and CCPP contained 1036.4 and 510.5 µmol Trolox/g dry material, respectively, while other polyphenol-rich powders exhibit values between 300 and 450 µmol Trolox/g dry material. Therefore, the antioxidant activity was higher with CCP addition in the present study, probably due to higher phenolic compound concentrations.

Furthermore, the highest antioxidant capacity among five ripe exotic fresh fruit was observed for camu-camu in another survey [28]. Camu-camu residue powders are a relevant source of vitamin C (8.2 mg/g of dried weight), and its extracts contain considerable phenolic content, even accounting for dehydration losses [3].

In the present study L* values were significantly different ($p < 0.05$) as a function of CCP concentrations, with values ranging from 49.11 (control) to 41.92 (5.0% CCP). However, no effects were observed concerning storage time or interactions with CCP concentrations. Similar to this study, another experiment noted no significant effects on L* values in refrigerated ground-lamb meat treated with camu-camu peel and seed extracts, with values ranging between 43.30 and 45.67 [25].

Fresh camu-camu residues presented an L* value of 51.8, increasing to 58.9 and 70.3, respectively, after dehydration by heat or freeze-drying [3], while L* values of 60.45 and 36.60 to 40.84 were observed for freeze-dried and dried camu-camu pulps, respectively [6].

During the storage period, and concerning CCP concentrations versus time, L* did not differ between the initial and final storage days, with a mean value of 44.75. Another survey observed that the luminosity value of vacuum-packed ground beef changed from 44.6 to 48.4 during 20 days of refrigerated storage [29].

Concerning the other investigated color characteristics, significant effects were observed, with decreases in a* values and increases in b* values as a function of CCP concentrations and decreases and increases in a* and b* values, respectively, as a function of time storage.

One survey demonstrated that peel and seed camu-camu extracts resulted in a* reductions from 11.13 to 6.51 and 15.60 to 6.71, respectively, in vacuum-packed ground

lamb meat stored at 4 °C for nine days. Decreases in b* values were also observed, with values ranging from 16.29 to 11.25 and 13.33 to 10.19 for peel and seed camu-camu extracts, respectively [25]. These findings corroborate with the present study, where decreased red hues (a*) in CCP-containing meat samples were observed. In addition, the same aforementioned study indicated that camu-camu peel and seed extracts led to a* value decrease, from 16.29 and 13.33 to 10.19 in vacuum-packed ground lamb meat stored at 4 °C for nine days [25]. It probably occurred due to color pigments oxidation by the action of radical species from lipid oxidation [25].

Concerning the microbial analyses, LAB did not present lag phase ($\lambda$), exhibiting growth since the beginning of the experiment. CCP addition did not interfere with the initial microorganism concentrations in the investigated meat samples. The control and CCP treatment values were 3.69 log CFU/g and between 3.58 and 3.85 log CFU/g respectively. Camu-camu powder addition did not inhibit LAB growth in meat, determined as 7.01, 6.98, and 7.60 log CFU/g in the treated group and 6.94 log CFU/g (control group).

On the other hand, CCP interfered in the kinetic behavior of LAB, resulting in decreased maximum growth rates, ranging from 0.45 to 0.35 log CFU/g/day for samples containing 3.5% and 5.0% CCP, respectively, and 0.49 log CFU/g/day in the control samples.

Some authors have reported the high antibacterial activity of alcoholic camu-camu pulp and seed extracts against *Streptococcus mutans* and *Streptococcus sanguinis*. These bacteria exhibit a high prevalence in the human oral cavity. Antibacterial activity was observed only for pulp extracts, with a minimum inhibitory concentration (MIC) of 62.5 μg/mL for the tested microorganisms [16]. It should be noted that *Streptococcus* belongs to the LAB group.

Another survey reported that camu-camu peel and seed extracts presented great antimicrobial activity against Gram-positive bacteria [4]. Other authors [3] reported MIC values ranging from 0.312 to 0.62 mg/mL for polyphenol-rich aqueous camu-camu extracts against *Staphylococcus aureus*. However, another assessment reported a lack of antibacterial effects of camu-camu seed and pulp extracts against *Staphylococcus aureus*, *Escherichia coli*, and *Saccharomyces cerevisiae* strains [30].

In the present study, slight decreases in initial PSY concentrations were observed in meat containing CCP with the value of 4.84 log CFU/g (5.0% CCP sample) and 5.092 log CFU/g in the control sample. CCP concentrations decreased the maximum PSY growth rate ($\mu_{max}$) to 0.788–0.912 log CFU/g/day in the CCP groups and 1.21 log CFU/g/day in the controls.

Another study found that dried and freeze-dried raw CCP extracts resulted in effects only against *Staphylococcus aureus*. In addition, the extracts displayed no antimicrobial activity against *Enterobacter aerogenes*, *Escherichia coli*, *Listeria monocytogenes*, *S. enterica* ser. Enteritidis, and *S. enterica* ser. Typhimurium [6]. These results corroborate the present study, where CCP did not inactivate *S. enterica* ser. Typhimurium concentrations in beef samples.

Strong antimicrobial activity of n-hexane and seed CCP extracts containing acylphloroglucinols and rhodomyrtone compounds against Gram-positive bacteria has been reported [4], with significant action against *Bacillus cereus*, *Micrococcus luteus*, *Staphylococcus epidermidis*, but no effects against Gram-negative bacteria such as *Pseudomonas aeruginosa* and *S. enterica* ser. Typhimurium. No CCP action against *Salmonella* and psychrotrophic bacteria was observed in the present study, probably due to the cell wall differences between Gram-positive and Gram-negative bacteria. However, effects against *Pseudomonas aeruginosa* were observed in a culture medium applying freeze-dried optimized camu-camu seed extracts [27]. Most studies have tested the antimicrobial effects of camu-camu extracts than CCP, and a few researches have analyzed the antimicrobial effects of CCP addition in meat, mainly in beef. Thus, we aimed to study only CCP effects to simplify camu-camu processing and use.

Regarding the pathogen behavior analyzed herein, *Salmonella* initial concentrations ($Y_0$) were significant ($p < 0.05$), with values ranging between 5.121 log CFU/g (control sample) and 5.046, 5.091, and 5.089 log CFU/g (treated samples). A fast decline in *Salmonella* concentrations with increasing CCP concentrations was observed, and it can be observed due to χ values of 26.9 and 40.32 in the 3.5% and 5.0% camu-camu addition treatments, respectively (Table 5), indicating more time for pathogen concentration decrease.

According to the shape parameter of the Weibull model, the 5.0% CCP addition (β = 0.359) caused more damage and stress to the evaluated pathogens than in the control group (β = 1.247) ($p < 0.05$). These shape parameters were 0.519 and 0.433 for the 2.0% and 3.5% CCP additions, respectively. Consequently, *Salmonella* exhibited higher adaptation in control group samples (0.0% CCP), demonstrating that camu-camu powder affected pathogen adaptability in ground beef.

One study using the micro-dilution method did not present any effects of dried and freeze-dried CCP extracts against *S. enterica* ser. Typhimurium and *Salmonella* Enteritidis [6], while another report indicated no antimicrobial activity of acylphloroglucinol and rhodomyrtone isolated from camu-camu peels and seeds against *S. enterica* ser. Typhimurium [4]. On the other hand, other authors [27] using the diffusion method on plate-cavity agar technic observed antimicrobial activity of freeze-dried optimized camu-camu seed extracts against *Salmonella* Enteritidis and *S. enterica* ser. Typhimurium, with 6.82 mm and 6.42 mm inhibition halos, respectively. *S. enterica* ser. Typhimurium and *S. enterica* ser. Enteritidis growth both were not inhibited by the action of freeze-dried and spray-dried CCP extracts in another survey [31]. Another assessment concluded that camu-camu seed and peel extracts had no effect against *S. enterica* ser. Typhimurium, concluding that these extracts exhibit strong action against Gram-positive bacteria, but little or no effect against Gram-negative bacteria such as *Salmonella* [4].

The results finding shows CCP addition did not decrease *Salmonella* concentrations in vacuum-packed ground beef, although it caused significant effects on some kinetic parameters.

## 5. Conclusions

CCP addition decreases lipid oxidation of vacuum-packed ground beef, although it affects color aspects, leading to a decreased red hue in the CCP-containing meat. Concerning antimicrobial activity, CCP does not interfere in *S. enterica* ser. Typhimurium behavior does not extend the *shelf-life* of vacuum-packed ground beef based on the concentration of certain spoilage microorganisms, acting only on the kinetic bacterial behavior parameters.

**Author Contributions:** Conceptualization, V.C., U.G.-B.; Data curation, V.C., U.G.-B.; Formal analysis, V.C., U.G.-B.; Funding acquisition, E.E.d.S.F., V.C., U.G.-B.; Investigation, J.L.d.S., J.M.L., V.C., U.G.-B.; Methodology, V.C., U.G.-B.; Project administration, V.C., U.G.-B.; Resources, V.C., J.M.L., U.G.-B.; Supervision, V.C., U.G.-B.; Validation, V.C., U.G.-B.; Visualization, J.L.d.S., V.C., E.E.d.S.F., U.G.-B.; Writing—original draft, J.L.d.S., E.E.d.S.F., V.C., U.G.-B.; Writing—review & editing, J.L.d.S., E.E.d.S.F. All authors have read and agreed to the published version of the manuscript.

**Funding:** This research was funded by the Foundation for Science and Technology (FCT, Portugal), Coordination for the Improvement of Higher Education Personnel (CAPES, Brazil) and Federal University of Mato Grosso, Brazil (UFMT).

**Institutional Review Board Statement:** Not applicable.

**Informed Consent Statement:** Not applicable.

**Acknowledgments:** The authors would like to thank CAPES (Coordination for the Improvement of Higher Education Personnel)—Brazil, for supporting the first author with a scholarship from the international Sandwich Exchange Program—PDSE 047/2017/Process no. 88881.189927/2018-01 and the National Council for Scientific and Technological Development—CNPq (Process: 310462/2018-5), and to PROPeq/PROPG-UFMT, Brazil. U. Gonzales-Barron and V. Cadavez are grateful to the Foundation for Science and Technology (FCT, Portugal) for financial support through national funds

FCT/MCTES to CIMO (UIDB/00690/2020). U. Gonzales-Barron acknowledges FCT, P.I., for the institutional scientific employment program contract.

**Conflicts of Interest:** The authors declare no conflict of interest.

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
