# Peer review of "Effects of Camu-Camu (Myrciaria dubia) Powder on the Physicochemical and Kinetic Parameters of Deteriorating Microorganisms and Salmonella enterica Subsp. enterica Serovar Typhimurium in Refrigerated Vacuum-Packed Ground Beef"

_agriculture, doi:10.3390/agriculture11030252_

Round 1

Reviewer 1 Report

Data on the quality of the developed models are essential as the quality of the fit affects the accuracy of the prediction, at least adding R2 and standard error of fit

Table 3, 4 i 5      put only symbol of asterisks without "2"

Line 93 add the type of cooling chamber

Line 130-131 It is not clear, it is necessary to link the information about incubation S. typhimurium at 8 °C until reaching the early stationary phase, determined with the aid of a previously constructed calibration curve, with the level of product contamination (5 log cfu/g).

Reviewer 2 Report

The paper is a study on the effects of camu-camu powder on the physicochemical and microbial quality of vacuum-packed ground beef. The paper is well written and detailed, and the authors have sufficiently motivated the importance of the study. The article would be valuable for the Meat Science domain. The authors have made a great effort to address each comment. They have provided more detail about their experimental design and the number of samples used and improved some language and style issues. The only minor change that I do not agree with is under Results, Table 1: Fewer decimals should be used for reporting the proximate analyses results – not more as it is a robust measurement, and the accuracy cannot be determined up to four decimals. Rather use 1 decimal
